# The Serious Illness Care Program in Oncology: Evidence, Real-World Implementation and Ongoing Barriers

**Safiya Karim [1,\*], Oren Levine [2] and Jessica Simon [1,3,4]**

[1]  Department of Oncology, Faculty of Medicine, University of Calgary, Calgary, AB T2N 4N2, Canada;
     jessica.simon@albertahealthservices.ca
[2]  Department of Oncology, Faculty of Health Sciences, McMaster University, Hamilton, ON L8V 5C2, Canada;
     levine.oren@gmail.com
[3]  Department of Community Health Services, Cumming School of Medicine, University of Calgary,
     Calgary, AB T2N 1N4, Canada
[4]  Department of Medicine, Cumming School of Medicine, University of Calgary, Calgary, AB T2N 4N2, Canada
\*  Correspondence: safiya.karim@ahs.ca; Tel.: +1-403-521-3166; Fax: +1-402-283-1651

**Abstract:** The Serious Illness Care Program (SICP), designed by Ariadne Labs, is a multicomponent intervention to improve conversations about values and goals for patients with a life-limiting illness. In oncology, implementation of the SICP achieved more, earlier, and better-quality conversations and reduced anxiety and depression among patients with advanced cancer. In this commentary, we describe the SICP, including results from the cluster-randomized trial, provide examples of real-world implementation of this program, and highlight ongoing challenges and barriers that are preventing widespread adoption of this intervention into routine practice. For the SICP to be successfully embedded into routine patient care, it will require significant effort, including ongoing leadership support and training opportunities, champions from all sectors of the interdisciplinary team, and adaptation of the program to a wider range of patients. Future research should also investigate how early conversations can be translated into personalized care plans for patients.

**Keywords:** serious illness care; palliative care; communication; oncology; goals of care; values and preferences

## 1. Why Are Serious Illness Conversations Needed?

Advances in treatment have expanded the spectrum of palliative-intent therapies for many cancers. Patients may now undergo multiple lines of treatment with the intent to slow the progression of their disease, control cancer-related symptoms, and ultimately improve their survival. As a result, patients are living longer with advanced cancer than ever before. Traditionally, palliative care was introduced to patients only at late stages of the cancer spectrum and was often mistaken as end-of-life care [1]. However, recent studies have highlighted the importance of early integration of palliative care alongside usual oncology care with several important benefits, including improved quality of life, reduced depression, and, in some cases, prolonged survival [2–6]. Specialized palliative care is a limited resource, and all oncology providers should have the skills to adopt an early palliative approach to care for patients with advanced cancer. This involves providing whole-person care, a focus on the quality of life, and mortality acknowledgment [7]. It also includes hearing and understanding each patient's goals and priorities for living with their serious illness.

Serious illness is defined as carrying a high risk of mortality within one year, having a strong negative influence on the quality of life and functioning, and is highly burdensome to a patient and his or her family [8]. Aggressive care for patients with serious illness often leads to worse quality of life and greater distress [9,10]. Early, high-quality goals of care conversations have been shown to have several benefits, including improved goal-concordant care and patient satisfaction, improved bereavement outcomes for caregivers,

and lower resource utilization [9–12]. Yet, conversations about care preferences tend to happen very late in the illness trajectory, often lack critical information, and tend to focus more on resuscitation preferences rather than eliciting patients' important hopes and fears that inform shared decision making [13,14].

Several barriers to conducting early, high-quality conversations have been cited in the literature. These include patient factors (e.g., anxiety, denial), clinician factors (e.g., lack of comfort, lack of training, fear of causing harm to patients), and system factors (e.g., limited time, lack of consistent electronic documentation, lack of clarity regarding which provider is most responsible for initiating the conversation) [15]. To address these challenges, Ariadne labs developed the Serious Illness Care Program (SICP), a multicomponent intervention to improve conversations about values and goals for patients with serious illnesses [16]. The goal of this program is for every seriously ill patient to have more, better, and earlier conversations with their clinicians about their goals, values, and priorities that will inform their future care [16]. The SICP has been studied and implemented in a variety of settings, including outpatient primary care [17,18], oncology [19–24], and hospitalized medical patients [25,26].

## 2. What Are the Components of the Serious Illness Care Program?

One of the major components of the SICP is a patient-centered conversation guide about priorities in the face of serious illness [16]. A checklist with patient-tested language has been developed to help structure a patient-centered conversation about priorities in the face of serious illness (Box 1). This guide focuses the conversation on goals and values, involves sharing prognosis according to individual information preferences, de-emphasizes treatment decisions, and builds a foundation for future values-based medical decisions. Other components of the SICP are listed in Box 2. System change methodology is used to adapt and implement the program into a given context. Clinician training is delivered through in-person or virtual workshops. Components of education include an evidentiary review, a demonstration with a standardized patient, orientation to the conversation guide, and a practical role-play session with qualified facilitators providing feedback. An important aspect of the program is a "trigger" for clinicians to remind them to conduct the conversation with the patient. In the original studies, triggering was performed via the research staff who would (1) email the clinician the day before the visit to notify them that the patient is due for the SIC conversation and (2) provide the clinician with the SICP materials on the day of the visit with the patient. Setting up the conversation and preparing patients in advance is also a key step. To this end, a pre-visit letter that introduces the topics of the serious illness conversation is provided to the patient. There is also a family communication guide for patients to share with their families after the discussion. Importantly the program encourages the use of standardized templates to record the conversation in the medical record where it can be readily retrieved for use in subsequent discussions or to provide context for in-the-moment decision making. Various SICP resources are available on the Ariadne Labs website [27].

**Box 1.** Serious Illness Conversation Checklist.

---

1. Set up the conversation
2. Assess understanding and preferences
3. Share prognosis
4. Explore key topics (i.e., goals, fears and worries, sources of strength)
5. Close the conversation
6. Document your conversation
7. Communicate with key clinicians

https://www.ariadnelabs.org/serious-illness-care/for-clinicians/
Source: Ariadne Labs. Serious Illness Conversation Guide. Available online: https://www.ariadnelabs.org/serious-illness-care/for-clinicians/ (accessed on 30 October 2021).

---

**Box 2.** Components of the Serious Illness Care Program (SICP).

1.  Population identification
2.  Training and coaching program
3.  Triggering
4.  Pre-visit letter
5.  Serious illness conversation guide
6.  Electronic medical record module documentation
7.  Family guide
8.  Implementation roadmap and system change resources

Source: Bernacki, R.; Hutchings, M.; Vick, J.; et al. Development of the Serious Illness Care Program: A randomised controlled trial of a palliative care communication intervention. *BMJ Open* **2015**, *5*, e009032.

## 3. Evidence for the Serious Illness Care Program in Oncology

The SICP was evaluated in the outpatient oncology setting in a single-center cluster-randomized trial from September 2012 to June 2016 [19]. Eligible patients were identified by clinicians' answers to the "Surprise question" (i.e., answering "No" to the question would you be surprised if this patient died within the next year?). The intervention included SICP tools, clinician training, and system changes. Control clinicians (medical doctor, nurse practitioner, or physician assistant) provided usual care to patients. The co-primary outcomes were goal concordant care and peacefulness at end-of-life, measured by a baseline survey and follow-up surveys every two months for two years or until death. A variety of patient-important secondary outcomes were studied, including quality of life, anxiety, depression, therapeutic alliance, and quality of death, among others [19].

A total of 91 clinicians and 278 patients were enrolled. A wide spectrum of cancer types was represented in the study population. Regarding feasibility outcomes, 98% of all clinicians felt the training was effective, and almost 90% followed through with at least one structured SIC conversation in the clinic. In terms of efficacy, there was no difference in the co-primary outcomes. Notably, these outcomes were only evaluable in 64 patients and were measured according to survey responses collected from patients near death or from family members early in bereavement. Collecting responses at these delicate times presented a challenge to investigators and was likely the reason for a limited response rate [22]. Analysis of secondary outcomes showed significant improvement in moderate to severe anxiety (10.2% vs. 5.0%; $p = 0.05$) and depressive symptoms (20.8% vs. 10.6%; $p = 0.04$) at 14 weeks. Improvements in anxiety, but not depression, were sustained at 24 weeks [22].

Importantly, the SICP intervention was associated with earlier, more frequent, and higher quality conversations [20]. Among the patients who died ($n = 161$), those in the intervention group were 17% more likely to have a documented conversation (96% vs. 79%, $p = 0.005$), and conversations occurred a median of 2.4 months earlier (143 days vs. 71 days, $p < 0.001$). Furthermore, clinicians using the conversation guide were far more likely to address key elements of the conversation, including illness understanding, prognosis, goals and values, and life-sustaining treatment preferences. Finally, significantly more intervention patients had documentation that was accessible in the electronic medical record (61% vs. 11%, $p < 0.001$) [20].

Several secondary analyses of the cluster-randomized trial have been conducted. In the first analysis, the aim was to assess concordance between written documentation and recorded audiotaped conversations and to evaluate adherence to the SIC conversation guide questions [21]. In a small sample of 25 audio recordings, most clinicians adhered closely to domains of the conversation guide, yet concordance between the content of the conversation and written documentation was low. Optimal written communication was achieved more often with the use of a standardized template, suggesting this might be a helpful component of quality improvement programs implementing the SICP. In the second analysis, a qualitative study of 25 SIC conversations was performed to characterize

the content of the conversations and identify areas for improvement [22]. The investigators found that the median duration of the conversation was 14 min (range 4–37 min) and clinicians spoke for approximately 50% of the time. Furthermore, patients were open to discussing challenging topics and often had clear preferences regarding their future care. However, clinicians struggled with offering a prognosis and responding to patients' emotions. Further training for clinicians to develop these skills was suggested as an area for further improvement.

Paladino et al. also conducted a secondary analysis of the randomized trial to determine the effect of the SICP on health care utilization at the end of life in oncology [28]. Health care utilization was abstracted from the electronic medical record (EMR) using the National Quality Forum (NQF)-endorsed indicators of aggressive cancer care at the end of life [29]. The study did not find any difference in the healthcare utilization between patients in the intervention and the control groups. Proposed explanations include that the intervention was not significant enough to translate early communication into changes in healthcare utilization, insufficient power to detect small but meaningful changes in healthcare utilization, and the validity and use of different measures to assess the aggressiveness of care (and associated expenditures) at the end of life. Interestingly, implementation of the SICP in the primary care setting did lead to decreased health expenditures at the end of life [30].

The results of the primary clinical trial and secondary analyses are summarized in Table 1.

**Table 1.** Published studies (primary and secondary analyses) of the Cluster Randomized Trial of the Serious Illness Care Program in Oncology.

| Author(s) | Type of Analysis | Participants/Sample | Primary Outcome (s) | Secondary Outcome (s) | Results |
|---|---|---|---|---|---|
| Bernacki et al. [22] | Primary analysis | $n$ = 278 patients (134 intervention, 144 control) $n$ = 91 clinicians (48 intervention, 43 control) | Goal concordant care and peacefulness at the end of life | Therapeutic alliance, anxiety, depression, survival | • No significant difference in the primary outcomes of goal concordant care or peacefulness <br> • Reduction in severe anxiety (10.2 vs. 5%, $p$ = 0.05), depressive symptoms (20.8% vs. 10.6%. $p$ = 0.04) at 14 weeks after baseline <br> • Anxiety reduction sustained at 24 weeks (10.4 vs. 4.2%, $p$ = 0.02) <br> • Depression reduction not sustained at 24 weeks (17.8% vs. 12.5%, $p$ = 0.31) <br> • No difference in survival or therapeutic alliance |
| Paladino et al. [23] | Secondary analysis | $n$ = 278 patients $n$ = 91 clinicians | N/A | Documentation of at least 1 serious illness conversation before death, timing of the initial conversation before death, quality of the conversations, accessibility in the EMR | • Higher proportion of patients had a documented discussion compared to controls (96% vs. 79%, $p$ = 0.005) <br> • Interventions took place earlier (median 143 vs. 71 days, $p$ < 0.001) <br> • More comprehensive conversations, greater focus on values/goals (89% vs. 44%, $p$ < 0.001), prognosis or illness understanding (91% vs. 48%, $p$ < 0.001), and life sustaining practices (63% vs. 32%, $p$ = 0.004) <br> • No difference in documentation about end-of-life planning <br> • More accessible documentation in the EMR (61% vs. 11%, $p$ < 0.001) |

**Table 1.** *Cont.*

| Author(s) | Type of Analysis | Participants/Sample | Primary Outcome (s) | Secondary Outcome (s) | Results |
|---|---|---|---|---|---|
| Geerse et al. [24] | Secondary analysis | 25 Audio recordings of 16 clinicians who conducted the serious illness conversation | Concordance between written documentation and recorded audiotape conversations, adherence to the Serious Illness Conversation Guide questions | N/A | • Documentation was concordant with the audio recordings 43% of the time<br>• 2 conversations (8%) were not documented<br>• Concordance was better when a standard template was used<br>• Clinicians addressed 87% of the conversation guide elements<br>• Prognosis was only discussed in 55% of patients |
| Geerse et al. [25] | Secondary analysis | 25 audio recorded serious illness conversations | Qualitative analysis to describe content of the conversation | N/A | • Median conversation duration = 14 min (range 4–37)<br>• 5 key themes: supportive dialogue between patients and clinicians, patients' openness to discussing emotionally challenging topics, patients' willingness to articulate preferences regarding life-sustaining treatments, clinicians difficulty in responding to emotional or ambiguous statements, challenges in discussing prognosis |
| Paladino et al. [26] | Secondary analysis | $n$ = 157 patients who died within 2 years of enrollment of the study (74 intervention, 83 control) | Mean number of aggressive indicators using National Quality Forum-endorsed indicators of aggressiveness at the end of life | Chemotherapy in last 14 days, $\geq$2 hospitalization or ED visits in last 30 days, $\geq$1 ICU stay in last 30 days, no hospice use or <3 days, death in acute care hospital | • Similar end of life healthcare utilization between intervention and control patients (0.9 vs. 0.9 aggressive indicators, $p$ = 0.84)<br>• Secondary outcomes showed no difference in proportion of patients with any aggressive care indicator (49% vs. 54%) |

## 4. Examples of Real-World Implementation of SICP in the Oncology Setting

Implementation of the SICP is occurring in real-world health care environments. Here, we describe several examples of quality improvement studies aiming to implement the SICP at various healthcare institutions since the publication of the cluster-randomized trial. At the Tom Baker Cancer Centre in Calgary, Canada, we implemented the SICP in two outpatient oncology clinics to improve communication and documentation amongst patients with advanced cancer [23]. The aims of this quality improvement initiative were to identify at least 24 patients (12 patients per clinic) for a SIC conversation and to document at least 95% of all conversations in the EMR. Two medical oncologists underwent a two-hour training session and were provided with the serious illness conversation guide [18]. In this study, each medical oncologist was asked to identify one patient per week (for a total of 16 weeks) that would be appropriate for a SIC conversation. This was based on the patient meeting one or more of the following criteria: (1) a response of "no" to the "Surprise Question", (2) any patient with a diagnosis of metastatic pancreatic cancer (due to the poor prognosis with a median survival of less than one year in this population), and/or (3) symptom scores of >7 (out of 10, where 10 is the worst possible score) on more than three categories on our patient-reported outcome dashboard. Once the patient was identified, they were provided with a pre-clinic visit letter, and the next clinic visit was booked as the last appointment of the day to allow adequate time to conduct the conversation. After the SIC conversation, the oncologist would document the conversation in a specific location in the EMR and fill out goals of the care order, if applicable. Within 48 h of the conversation, the primary clinic nurse would conduct a post-survey conversation with the patient that aimed to understand the patient experience. In addition, each oncologist filled out a post-conversation survey to rate their comfort level in conducting the SIC conversation.

While the results of this study did not meet its primary endpoint of conducting 24 SIC conversations, >95% of the conversations that did take place were documented in the EMR. Furthermore, baseline rates of documentation improved for both oncologists (4.2–5.4% of all patients for clinician A and 0–7.3% of all patients for clinician B over a 16 week period).

In another quality improvement initiative conducted at the Abramson Cancer Center of the University of Pennsylvania, Kumar et al. sought to understand the experience and perceptions of patients who had a SIC conversation with their oncology clinician [24]. Eligible patients were administered a five-item survey that was adapted from the instrument used in the original cluster randomized trial. Responses were recorded using a Likert scale, where 1 = decreased a lot and 7 = increased a lot. In addition, patients were also asked to evaluate the overall worth of the conversation, where 1 = not worthwhile at all and 4 = extremely worthwhile. In total, 32 patients agreed to participate in the study, and 31 completed all survey questions. After the SIC conversation, 55% of all participants reported increased understanding of their future health, 42% reported an increased sense of control over future medical decisions, 58% reported increased closeness with their physician, and 42% reported increased hopefulness about quality of life. Many patients reported no change in these domains, and a small proportion of participants (<10%) reported a negative reaction to the conversation. Overall, 90% of participants believed it was worthwhile to talk to their clinician.

Implementation of the SICP at the Juravinski Cancer Centre in Hamilton, Ontario, is currently underway. This is a 3-phase program aimed at (1) delivering virtual clinician training on the SICP, (2) implementing the SICP with a pilot group of health care providers within the breast, lung, and central nervous system disease site groups, and (3) scaling and spreading the implementation of the SICP throughout all disease site groups by 2023. The virtual training program will be evaluated through pre- and post-training questionnaires to assess knowledge, skills, and confidence with conducting the SIC as well as any perceived barriers. During the implementation phase, Plan-Do-Study-Act (PDSA) cycles will be used to evaluate and adapt the delivery of the SICP. In addition, a number of strategies to ensure the long-term sustainability of this project have been considered, including a commitment by operational leadership, the development of a SICP dashboard to monitor performance, and ongoing training for medical and radiation oncology residents as well as new oncologists.

## 5. Challenges and Barriers to Real-World Implementation

The SICP was developed as a multicomponent intervention with a set of tools and processes designed to address barriers in conducting conversations about prognosis, values, and goals with patients. However, there are ongoing patient, provider, and system-level barriers that are affecting the widespread uptake of this initiative.

Our experiences of the challenges to implementing the SICP are similar to other advance care planning initiatives [31] and can be understood using Michie's capability, opportunity, motivation, behavior (COM-B) model [32]. (Figure 1). Physical and psychological capability is addressed within the SICP with initial training for clinicians, and the checklist for the conversation using patient-tested language. Patients are prepared ahead of meeting the clinician using a pre-conversation letter. However, as noted in the study by Geerse et al. [22], clinicians still struggle with discussing prognosis and responding to patients' emotions, and education beyond the initial training workshop may be needed to enhance these aspects of communication skills.

Capability is also enhanced through automatic triggers in the EMR to identify appropriate patients. There has been an interest in using algorithms to identify patients at high risk of mortality and/or behavioral nudges to help clinicians initiate more SIC conversations [33–36]. In a step-wedged randomized clinical trial of 78 clinicians and 14,607 patients, Manz et al. showed that the combination of machine learning mortality predictions with behavioral nudges to oncology clinicians significantly improved the rate of SIC conversations [36]. The intervention led to a 4-fold increase (3.6–15.2%) in SIC

conversations among patients with the highest risk of 180-day mortality. A number of additional important behavioral change methods were included in the study design study, such as peer comparison, performance feedback, and opt-out email and text reminders.

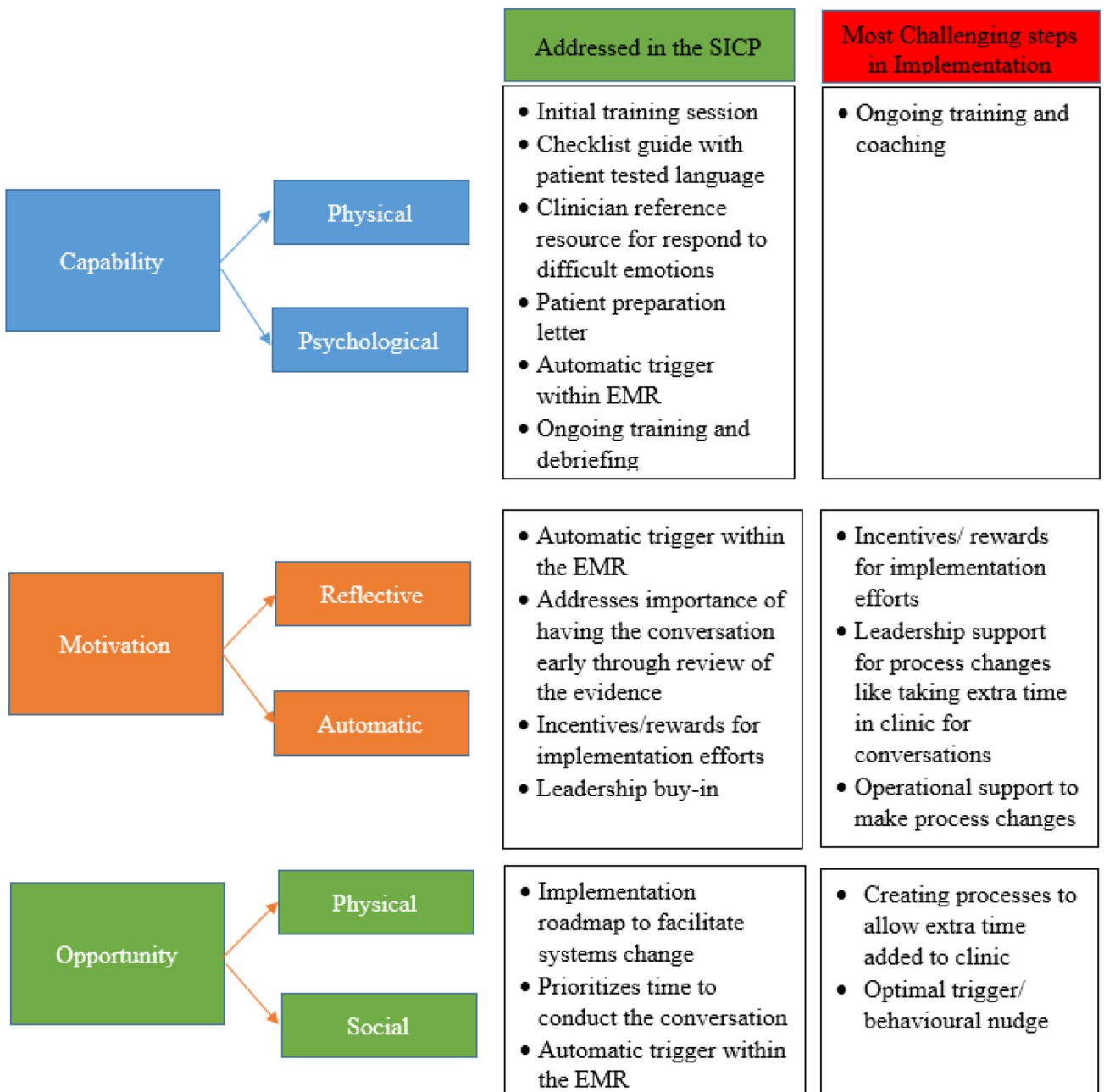

**Figure 1.** COM-B model components and the SICP.

The SICP addresses motivation in various ways. There is an emphasis on the need for early conversations, thereby providing reflective motivation for providers. Automatic triggers in the EMR also provide motivation to clinicians to initiate conversations with certain patients. Cultural adaptation of the guide has occurred to facilitate cultural safety and enhance patient and/or provider motivation to participate in conversations [37].

Opportunity is often the greatest challenge. The SICP prioritizes time to conduct the conversation and provides the automatic trigger to clinicians. However, if processes are not adjusted to allow adequate opportunity (i.e., dedicated clinic time for SICP conversations), uptake in the long-term will likely dwindle. The SICP implementation roadmap advocates

that leaders should be engaged early to support change. In addition, to accommodate the time needed for conversations, clinic processes usually require modification. Therefore, in addition to leadership buy-in, operational staff must work together to create the opportunity and accommodate the processes to support the integration of SICP into practice.

## 6. Future Directions

The SICP aims to improve conversations about values and goals for patients with serious illnesses. In oncology, the pivotal randomized trial showed the implementation of the program was associated with earlier, more frequent, and higher quality conversations and reduced anxiety and depression in patients, compared to usual care. However, for the clinical trial findings to be replicated and widely adopted in routine practice, several ongoing barriers need to be addressed. First, ongoing training and coaching of clinicians are necessary to allow long-term capability. In British Columbia (BC), Canada, the BC Centre for Palliative Care established an online Community of Practice where facilitators, trainers, and coaches can connect and share their experiences and lessons learned [38]. These types of initiatives developed locally can support the sustainability of the SICP. Furthermore, SIC communication training should be embedded in oncology residency education. With a shift to competency-based medical education [39], this can be prioritized as a mandatory training experience for all future clinicians.

Second, implementation of the SICP requires interdisciplinary engagement, including champions from various areas, including nursing, social work, information technology, and administration. This requires coordination and clarity of roles as well as potentially significant adaptations of current EMRs and clinic processes. Behavior change among clinicians is unlikely without significant effort for system changes to support the implementation of the SICP.

Finally, attention to patient–provider power differentials and cultural safety is paramount in goals and values conversations. The BC health authority has translated the SIC guide into several different languages, and they continue to work with First Nations communities to meet the unique language and cultural needs of this population [38]. Further adaptation of the program for a variety of other cultures and marginalized populations [40] or to address the needs of geographically remote populations will require ongoing efforts and engagement from varied stakeholders. Eventually, culturally appropriate conversations should translate into personalized care plans for patients and their families.

In summary, the SICP is an excellent intervention to improve the care of patients with advanced cancer. However, without ongoing training opportunities, interdisciplinary champions, and attention to equity and diversity, uptake and implementation of the SICP in routine practice may continue to be slow and effortful.

**Author Contributions:** Conceptualization, S.K., O.L. and J.S.; methodology, S.K. and O.L.; writing—original draft preparation, S.K. and O.L.; writing—review and editing, S.K., O.L. and J.S. All authors have read and agreed to the published version of the manuscript.

**Funding:** This research received no external funding.

**Institutional Review Board Statement:** Not applicable.

**Informed Consent Statement:** Not applicable.

**Data Availability Statement:** Not applicable.

**Conflicts of Interest:** The authors declare no conflict of interest.

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
