# Peer review of "The Serious Illness Care Program in Oncology: Evidence, Real-World Implementation and Ongoing Barriers"

_curroncol, doi:10.3390/curroncol29030128_

Round 1

Reviewer 1 Report

The authors have now properly addressed my concerns

This manuscript is a resubmission of an earlier submission. The following is a list of the peer review reports and author responses from that submission.

Round 1

Reviewer 1 Report

dear authors, this is a very interesting manuscript. 

the introduction will benefit from a minor revision to include some details on serious illness treatment programs or serious illness care program. 

please provide a review (and supporting evidence) on how exploring patients' aims and values with physicians and critically sick patients may generate high-value healthcare, increasing patient outcomes and lowering expenditure. 

see for example programs aimed at increasing, earlier, and better critical illness communication have the potential to save money you please refer to https://pubmed.ncbi.nlm.nih.gov/31904304/ this maybe come after reference 22. 

abstract will benefit from the conclusion and recommend further research -  perhaps future research should concentrate on the mechanisms that lead to tailored treatment plans through communication regarding patients' prognosis, values, and objectives.

Reviewer 2 Report

  1. The Serious Illness Care Program (SICP), designed by Ariadne labs was described in this commentary. It included why are serious illness conversations needed, what are the components of the serious illness care program, evidence for the serious illness care program in oncology, examples of real-world implementation of SICP in the oncology setting, and challenges and barriers to real-world implementation. This commentary is worth to be implemented widely to improve the quality of life for patients with serious illness.
  2. The writing format of references need to be consistent. e.g 30.

Reviewer 3 Report

Although interventions such as SICP are probably useful, especially among oncologists with otherwise no substantial training in the field of communication, the available data is in my opinion not yet sufficient as creating a control group for such analysis is very challenging (most of the presented studies have flawed methodology). Is there any study comparing SICP with similar intervention? Nevertheless, highlighting the importance of interventions as such is valuable. Therefore, this paper may be suitable for publication after certain adjustments. I have the following comments. 

I suggest the authors to add a comprehensive figure which summarizes advantages and pitfalls of this method (or its implementation). In line with these, I suggest authors to abbreviate the story telling when describing studies focusing on the most important parts (results and their implications), whereas advantages and pitfalls should be described in more details.

I suggest the authors to make a comprehensive table which will include all the performed clinical studies on this topic (number of patients, population, design, outcome).

The authors should offer some future directions in which they will among others provide reader with putative ways for implementing this method in real life clinical settings. In fact, this should be the most important aspect of this commentary hence I suggest authors to properly address this issue.